# Transparency and E-Government in Electronic Public Procurement as Sustainable Development

Jorge Hochstetter [1], Felipe Vásquez [1], Mauricio Diéguez [1,*], Ana Bustamante [1] and Jeferson Arango-López [2]

1   Department of Computer Science and Informatics, Universidad de La Frontera, Temuco 4811230, Chile
2   Systems and Informatics Department, Universidad de Caldas, Manizales 170001, Colombia
*   Correspondence: mauricio.dieguez@ufrontera.cl

**Abstract:** The transparency of electronic procedures has become an important strategy to reduce corruption within state organizations and thus promote the sustainable and efficient management of fiscal resources, vital elements in the development of a country. E-government processes have become an important line of development, in which substantial investments have been made to have processes that allow for transparency in a large part of the country's activities, specifically in the contracting and purchasing of public properties and services. The objective of the study is to present an overview of the work on initiatives that have been used around transparency and electronic procedures of electronic governments to identify which of these initiatives are associated with transparency and which effectively apply to electronic procedures for transparency to learn how these procedures allow for sustainable development of governments. The methodology used in this work was a systematic mapping of the literature, and the main findings suggest that this is a little-explored area.

**Keywords:** E-government transparency; electronic procedures; sustainability transparency





## 1. Introduction

One of the elements that has had a substantial impact on the credibility of state institutions in the population is the acts of corruption that affect some public administration processes [1]. The corruption perception index in the public sector [2], out of a total of 180 countries and territories evaluated, indicates that two-thirds of these do not exceed 50 points on a scale of 0 to 100 points, where 0 corresponds to very corrupt and 100 to very clean, which means that there are severe problems of corruption. The observed average corresponds to 43 points. While it is true that this same study indicates that corruption levels have not increased since the last observation, the data indicate that corruption levels are still high [2].

To counteract this effect, laws have been enacted requiring public entities to make their procedures more transparent, but compliance with these regulations still needs to improve [3].

One of the main allies to increase government transparency has been implementing and using information technologies for E-government [4–6]. In different countries, these initiatives are supported and regulated by laws on using information technology in the public sector [1]. Thus, governments use electronic procurement and contracting systems to achieve a transparent process in acquiring properties or services and hiring personnel, often linked to procurement and contracting web portals [6,7]. As pointed out by several authors [1,8–10], these systems increase active transparency and improve relations with suppliers. Thus, it is expected that, as a consequence, they will generate a decrease in corruption practices, an increase in trust in public institutions, an increase in the productivity of procurement activities, the attraction of new suppliers, and an increase in the interest of citizens in public calls for tenders for the hiring of personnel.

Governments consider transparency a vital method of public liability and responsibility towards the community and a critical element of good governance [7]. Thus, there are studies on transparency practices and the sustainability of public policies, considering that there is direct contact with citizens and the large amounts of economic resources managed by government entities [7,8].

On the other hand, the United Nations (UN), in September 2015, established 17 sustainable development goals and 169 targets that have an integrated and indivisible character. These goals are defined as global and their application is universal, considering each country's capacities and levels of development and respecting their policies and priorities [11]. Although the objectives are global, each government will establish its local objectives, considering the country's circumstances. Furthermore, each government will decide how to incorporate global goals into national policies and strategies. Therefore, it is crucial to understand the relationship between sustainable development and other economic, social, or environmental processes [11].

In this sense, the 2030 Agenda for Sustainable Development (UN) highlights transparency as a valuable tool to fight corruption and promote sustainable development of countries, for example, by stating that institutions must be transparent and effective in their accountability [9]. Since trust in government and the public sector is related to society's assessment of the credibility, fairness, competence, and transparency of political institutions, several authors [8,10–12] report a positive relationship between transparency, low corruption, and trust in government. Therefore, the aim is to open the government to the citizens [4], generating an agenda of reforms in favor of transparency, accountability, and collective responsibility [5]. Thus, several countries have enacted laws requiring government entities to publish information on bidding processes for properties, services, and personal contracts (electronic procedures) [13]. Electronic public procedures mediated by E-government systems play an essential role in national government programs since they aim to enable active transparency and efficiently favor relations with suppliers [14].

Thus, the importance given by citizens and the constant pressure toward higher levels of transparency make this a relevant problem for public agencies and a challenge for the entire state. It is expected that the public electronic procedure systems will contribute to the increase in transparency, which would bring about possible decreases in corruption [8,11,15,16], increased confidence in the procurement system, and increased productivity of both procurement and service activities, in addition to attracting new suppliers to negotiate with the government [17].

Both a correct definition and implementation, together with high levels of compliance with transparency policies concerning public procedures, can increase the economic and social benefit of public procurement or contracting, as both processes are often subject to risks such as (i) lack of technical knowledge of the organization that prepares the bases for purchases or contracts [13], (ii) corruption, that has a harmful impact by, artificially, raising the final price of the product [18], (iii) self-removal of competitive providers due to the belief in pre-agreed bids [13,19], (iv) the existence of formalization bids, i.e., processes that are created to cover services that have been rendered without initially going through an official procedure, which reinforces the perception that the formal system is a bureaucratic calibrated procedure [6,20], (v) the use of competitive bidding for public contracts that in reality seek to regularize "contract" positions, which reinforces the belief that the competition may be rigged [13].

Despite these expectations, one of the areas in which there is a more excellent perception of corruption is the bidding processes for properties and services and calls for public tenders for personnel recruitment. Therefore, it is most urgent to make them transparent.

From the above and considering studies on transparency, which state purchases, contracting, and acts on third parties, and the matters that show the worst results in audits [18], the need arises to know the transparency policy initiatives in electronic public procurement within E-government. In addition, there is a discussion in the scientific community about the real effects of the inclusion of E-government in the fight against

corruption, indicating that there needs to be a clear organization regarding E-government initiatives and corruption [17]. Then, it is necessary to know the advances in implementing E-government measures in public administration, what initiatives are proposed by the scientific community, and the formalization of these initiatives. For this purpose, the systematic mapping methodology proposed by Petersen [21] was applied based on the work of systematic reviews proposed by [20]. The authors adapted this methodology from the area of medical research to use information technology (IT). The study's objective is to present, through a general review of public electronic procurement, what is associated with transparency and generates proposals for the transparency of e-proceedings.

Our contribution consists of identifying proposals seeking to implement improvements in transparency within E-government programs. Although countries promote laws associated with transparency that oblige public entities to make their procedures more transparent, the level of compliance with these regulations still needs to be higher [13,22]. Therefore, this study aims to map their advance and the proposal's maturity.

This presents an opportunity to define a formal and systematic process to support organizations in defining, planning, implementing, and monitoring the transparency activities to which they should be subject [23].

Our work is structured as follows: in Section 2, we described the electronic public procurement. In Section 3, we present the methodology used to create our proposal. Section 4 presents the SMS results. Section 5 includes the results-based discussion, followed by Section 6 describing the study's limitations. In Section 7, we describe the conclusions and the future work associated with the investigation.

## 2. Public Electronic Procedures

In public bidding processes for economic consumption, as in public tenders for personnel recruitment, there is a client and, in both cases, a government entity.

In the following, we will explain the public bidding processes of this electronic procurement. In Figure 1, we illustrate the process using BPMN graphical notations. First, we identify the actors according to their roles; two lanes represent them: the lower lane corresponds to the supplier and the lower lane to the customer. The empty circles represent the beginning of the procedure. The light red circles represent the end of a process. The clocks represent a deadline or a constraint; these should be noted after a process. The diamonds correspond to decision-making after a process. Rectangles correspond to sub-processes, and flows represent information messages or sequence controls. These multiple interactions are managed through electronic platforms.

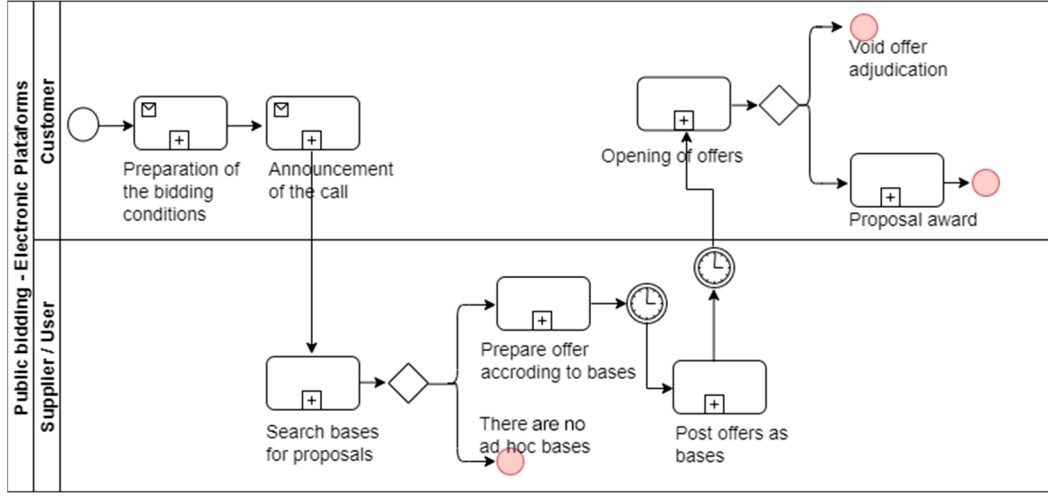

**Figure 1.** Public bidding processes for electronic public procurement.

The electronic public procurement processes include the following stages:

**Preparation of the bidding conditions**: First, the client or purchaser publishes the call for proposals or public bidding and classifies it by topics and areas. Next, the client defines the judge principle and alternative system for the proposals or bidders. This step needs to specify terms, price range, and characteristics of the good or service required or the characteristics of the position.

**Call announcement**: The public bid will be shown on an e-platform to request bids from providers registered on the platform and those who wish to apply for a public position.

All public tenders, such as calls for public positions managed through electronic platforms, are, by definition, open, i.e., any user registered on the platform may submit a bid or apply for a position.

The platforms support a database of suppliers, which is also classified by areas and subjects of specialization.

**Search bases for proposals**: The platform then sends an email to any supplier whose expertise matches that of the new bid (customer-to-supplier flow). In addition, suppliers can perform a manual search for public tenders at any time according to a set of predefined criteria (proposal search bases). This is different for public tenders, where users must check the web platform for tenders.

**Prepare offer according to bases**: Once the supplier has decided to enter a public bidding process, the procurer prepares the proposal according to the specifications of the bidding document (prepare bids). During this stage, some public bidding processes allow suppliers to send questions to the client, who can respond publicly to all suppliers at once through the platform or privately via email. In addition, the client can organize a public expression of interest meeting, where the client and bidders can meet to clarify the problem the project intends to solve. However, no consultation sessions are held for users for calls for tenders.

**Opening of offers**: At this phase, the client reviews all bids. In general, a governance institution has internal processes to rank the scores of supplier bids or user resumes.

**Proposal award**: The name of the supplier awarded the proposal is published on the technology platform; when no supplier meets the client's specifications, the procedure is declared void. The client provides an official contact and contacts the supplier to coordinate the purchase. In the case of a public tender, the user is contacted by email.

Achieving a transparency policy for electronic procurement demands exertion above active transparency (government agencies must publicize helpful, punctual, and pertinent information on their websites). First, however, a culture is needed in the organization, of which transparency should be an elementary bastion [24].

### 3. Research Method

PRISMA's systematic mapping study (SMS) [25] technique offers a way of verifying, analyzing, and categorizing results related to a specific topic or area of interest, thus making it possible to determine the scope of the research and classify the knowledge obtained.

Conducting a systematic mapping study involves following or sequentially adapting the stages described in Figure 2. Thus, as the different stages of the process are completed, concrete results are obtained that form the direct input for the next step to achieve a systematic mapping as a result.

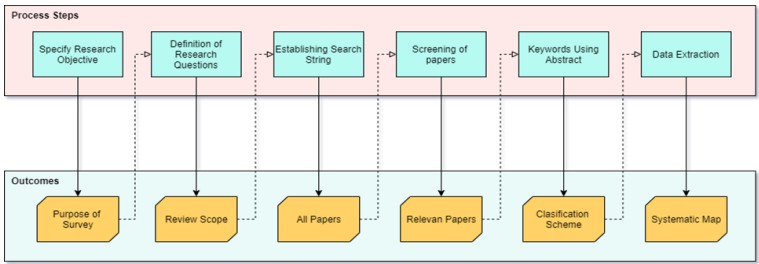

**Figure 2.** Stages of the systematic mapping process, reprinted from Vásquez (2021) [26].

One of the most notable differences between a systematic review and the methodology used in this paper is that the report generated by the mapping mainly seeks to catalog and classify the existing evidence in the literature, identifying knowledge gaps. At the same time, the systematic review also seeks to perform qualitative or quantitative syntheses in addition to a more detailed narrative of the papers [27].

Furthermore, according to Petersen [21], the mapping only verifies the abstract, results, and conclusions, while the review performs an exhaustive analysis of all the documents found [27].

The activities that make up the systematic mapping process are described in the following sections.

### 3.1. Goal and Research Questions

This SMS aims to identify and classify the initiatives used around transparency and E-government in electronic procurement.

For our SMS, the process started with the formulation of research questions, which is the backbone of the mapping because it supports an overview of a given area [21]. Table 1 presents each of the questions and their motivation. Through these, it was possible to select, analyze, and categorize the information found in the study area.

**Table 1.** Research Questions to be applied.

| Research Question | Motivation |
| --- | --- |
| RQ1: How many selected papers put forward proposals related to transparency and E-government policies? | Recognizing the documents that put forward proposals related to E-government transparency initiatives is the first step to being able to answer the following questions. |
| RQ2: How many papers on transparency and E-government deal with electronic procedures? | Electronic procedures play an essential role in transparency matters; recognizing how many are related helps to categorize the different levels of support from electronic procedures to transparency. |
| RQ3: How have proposals related to transparency and E-government in electronic procedures evolved over the years? | It allows the analysis of proposals for the use of transparency and E-government in electronic procedures and their use over time. |
| RQ4: In what type of journals are the selected papers being published? | It seeks to represent part of the relevance of the different proposals and their value within the scientific community. |
| RQ5: How many papers on transparency and E-government in electronic procedures address the sustainable development of governments? | Sustainable development must be supported by initiatives or policies of transparency and E-government in electronic procedures; knowing the amount of work focused on it becomes relevant to the state of adoption of sustainable development. |

### 3.2. Search String

For the construction of the search string, keywords were identified from the research questions and the study's objectives. Subsequently, they were ordered and concatenated through logical connectors. Finally, the chain obtained was reviewed and validated by the researchers. The resulting line was: ("Transparency" AND ("government" OR "E-government" OR "electronic government") AND ("procurement" OR "electronic procurement" OR "e-procurement" OR "recruitment" OR "electronic recruitment" OR "e-recruitment" OR "electronic service" OR "e-service" OR "acquisition" OR "electronic acquisition" OR "e-acquisition" OR "purchase" OR "electronic purchase" OR "e-purchase" OR "hiring" OR "electronic hiring" OR "e-hiring" OR "call for tender" OR "bidding")).

### 3.3. Data Extraction

Databases and websites with access to digital libraries were included in the search and data extraction process. They contain search engines that allow searches using search strings to download many related papers. The data sources selected were WoS, Scopus, Springer, Wiley, and ScienceDirect.

### 3.4. Inclusion and Exclusion Criteria

The selection of the studies found through the academic search engines mentioned above was based on the following inclusion/exclusion criteria:

Inclusion criteria:

- Papers published in English from journals and conferences.
- Full papers related to transparency and E-government in electronic procurement.
- Papers from 2012 onwards.

Exclusion criteria:

- Technical reports, abstracts, editors' comments, state the of art.
- Studies prior to 2012.
- Studies without an author.
- Documents that do not include the use of classrooms in the context of transparency and E-government in electronic procurement.
- Duplicate studies in different databases.
- Documents that do not come from traceable journals or procedures.

### 3.5. Search Execution

The search string was applied to the selected sources, and an initial quantity of 705 jobs was obtained (see Table 2). Then, the information was extracted using the export tools of each digital library. After eliminating doubly indexed jobs, the number was reduced to 437 jobs.

**Table 2.** Sources for the automated search.

| Electronic Data Source | Results |
|---|---|
| Web of Science | 151 |
| Scopus | 428 |
| Springer | 51 |
| ScienceDirect | 68 |
| Wiley | 7 |

Once the duplicates were eliminated, the exclusion/inclusion criteria were applied, for which three of the authors achieved an agreement on the Cohen kappa coefficient of 0.89, which is considered acceptable [28]. This coefficient was achieved by randomly selecting 20 papers that the reviewers reviewed for agreement. Next, each of the papers was marked with the categories YES/NO/DOUBT, leaving the latter for group discussion, first through the abstract, leaving 110 papers selected, and then through the reading of results and conclusions, filtered to the final count of 71 (see summary in Figure 3).

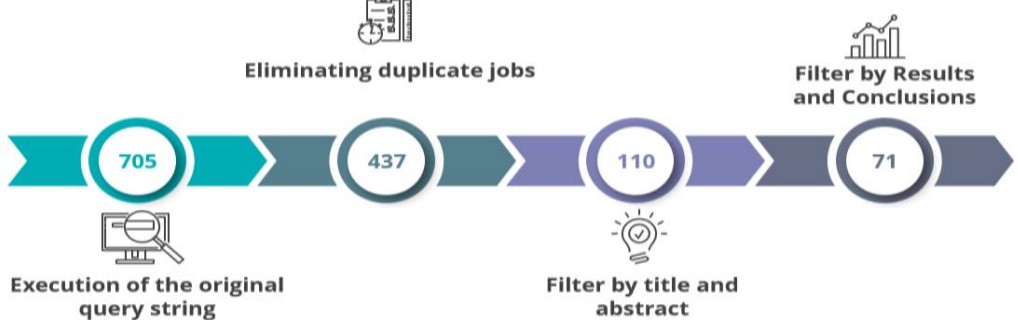

**Figure 3.** Search Execution and Filtering Summary.

### 3.6. Classification Scheme

Classification scheme: For the classification of publications, a scheme with three dimensions was generated: temporal, type of proposal, and sub-area. In the temporal dimension, the papers are classified according to the year in which they were published, considering the last ten years, 2012–2022.

The sub-areas are corruption, transparency, public procurement, maturity model, E-government, ICTs, legal standards, and accountability.

Although some papers could be classified into more than one category, the construction of these categories was based on a combination of characteristics to minimize this problem.

The proposal type dimension classifies the papers into:

- Analysis papers that describe analyses, comparisons, or reviews of the literature on transparency in electronic procurement.
- Use studies or works related to transparency in electronic procurement for subsequent application.
- Implementation: proposal of solutions aimed at transparency in electronic procurement.

*3.7. Map Construction*

The final result of the systematic mapping was a map to simplify representation and analysis. Figure 4 represents the papers categorized into implementation, use, and analysis. The right-hand part of the figure represents the publications' rankings by year ranges.

**Figure 4.** Representation of the systematic mapping.

## 4. Results

Transparency and E-government policies are of utmost importance, as they aim to achieve efficiency in the public management of governments. Consequently, knowing the transparency policies and initiatives in E-government is challenging due to the complexity of integrating different socio-cultural theories.

This paper aims to describe transparency policy initiatives in electronic public procurement and their contribution to the levels of sustainable development of E-government.

The resulting mapping (Figure 4) was constructed from 71 selected papers. Table A1 represents all selected papers, the different dissemination channels, publication year, and their citation count. According to this, it was possible to obtain data from the principal dissemination channels in this study area. There is an increase in the number of papers that analyze, compare, or review the literature on transparency in electronic procurement, transparency, and corruption.

It was identified that 86% of the works included analyses, comparisons, or literature reviews regarding transparency in electronic procurement from 2012–2022. In the same period, 11.56% of the works proposed solutions, and only 2.01% of the pieces used studies or works related to transparency in electronic procurement for their subsequent application. Concerning the latter two percentages, a possible explanation is that establishing proposals for transparency in electronic procurement requires methodological guidelines and cultural changes, making it a complex process to follow, which is a challenging area. According to what we have been able to review from the technical literature, we are facing a little-explored area. The following are answers to the research questions posed above.

### 4.1. RQ1: How Many Selected Papers Put Forward Proposals Related to Transparency and E-Government Policies?

Of the 71 papers found, we identified 55 papers that address transparency as their central theme, and among these papers, seven focus on transparency in the context of E-government. We have classified these seven papers under a common objective: assess the status of e-procurement implementation for government procurements. For this objective, we have identified five methodology types: (i) case study. We have classified in this category those works that indicated this methodology. (ii) Survey—descriptive statistics. We used the same criteria as above for this category. (iii) Theoretical argumentation. We classified the described papers using a rational sequence of facts and existing theories to support their proposals. (iv) Qualitative content analysis. We classified qualitative content analysis as those papers that used descriptive statistics to show results. (v) Quantitative analysis. We classified as quantitative analysis those papers that reported user some inferential statistical approach. Each paper presents a particular proposal, as shown in Table 3.

**Table 3.** Proposed solutions for transparency in E-government.

| Goal | ACM | IEEE | EDP Sciences | MDPI | Iopscience | Elsevier |
|---|---|---|---|---|---|---|
| To improve public e-procurement and transparency | [29] | [3,16] | [30] | [31] | [32] | [14] |
| **Methodology** | | | | | | |
| Case study | | | | | | [14] |
| Theoretical argumentation | | [3,16] | | | [32] | |
| Quantitative analysis | [29] | | [30] | [31] | | |
| Qualitative analysis | | | | [31] | | |
| **Proposals** | | | | | | |
| E-Government Procurement Observatory Maturity Model (eGPO-MM) | | | | | | [14] |

**Table 3.** *Cont.*

| Goal | ACM | IEEE | EDP Sciences | MDPI | Iopscience | Elsevier |
|---|---|---|---|---|---|---|
| Maturity model as a tool to measure tendering transparency when government agencies procure software development | | [16] | | | | |
| Ontology-based text mining and clustering techniques were applied to automatically identify and classify products | [29] | | | | | |
| Design of a maturity model for public procurement processes | | [3] | | | | |
| Method of data collection based on crowdsourcing that allows generating datasets of public procurement processes from the unstructured information published on the web | | | [30] | | | |
| Techniques for detecting disruptions in a set of open and disparate data integrated into a knowledge graph, which includes tender, company, and expenditure data, through a platform (TheyBuyForYou) based on linked data | | | | [31] | | |
| The development of a procurement data standard tool to promote transparency for compliance monitoring of the procurement process in all government agencies | | | | | [32] | |

### 4.2. RQ2: How Many Papers on Transparency and E-Government Deal with Electronic Procedures?

We found 71 papers related to transparency and E-government policies, of which we identified that 63 included analyses, comparisons, or reviews of the literature on the subject; only two papers included other studies or works for their subsequent application; and six reports addressed solutions oriented to electronic procurement. Table 4 presents a summary of these papers.

**Table 4.** Works that address electronic procurement.

| Goal | IEEE | Inder Sciences | MDPI | HeinOnline | Other |
|---|---|---|---|---|---|
| Assess the status of e-procurement implementation for government procurements | [3,16] | [33] | [31] | [34] | [35] |
| **Methodology** | | | | | |
| Case study | | | | | |
| Survey—descriptive statistics | | [33] | | | |
| Theoretical argumentation | [3,16] | | | [34] | [35] |
| Quantitative analysis | | | [31] | | |
| Qualitative analysis | | | [31] | | |
| **Proposed Theories** | | | | | |
| Evaluate the e-readiness level and utilization of the e-procurement system by a state | | [33] | | | |
| Measure tendering transparency when government agencies procure software development | [16] | | | | |
| Examine the rationale for E-government procurement provisions in bilateral trade agreements and highlight the benefits of using electronic systems | | | | | [35] |
| Measure transparency of tendering procurement processes | [3] | | | | |
| An analysis is carried out on e-administration, its objectives, and how it is developed and manifested in government procurement | | | | [34] | |
| Techniques for detecting disruptions in a set of open and disparate data integrated into a knowledge graph, which includes tender, company, and expenditure data, through a platform (TheyBuyForYou) based on linked data | | | [31] | | |

### 4.3. RQ3: How Have Proposals Related to Transparency and E-Government in Electronic Procedures Evolved over the Years?

In Figure 5, we can see an increase in the last few years in papers related to transparency in electronic public procurement and E-government. The year 2022 is a natural drop if we consider the review times of papers to be published.

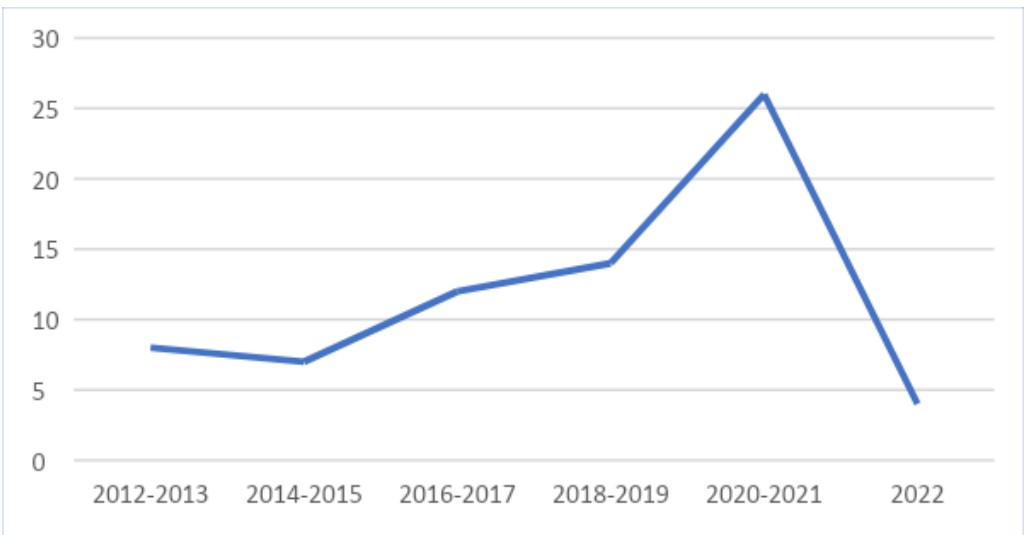

**Figure 5.** Number of documents by publication year.

The analyzed information was processed in an Excel database, with which the research questions could be answered.

*4.4. RQ4: In What Type of Journals Are the Selected Papers Being Published?*

In order to identify the number of papers by type of publication, i.e., journal paper, conference, and book chapter, from the bibliographic database from which they were extracted, a bar chart was made (see Figure 6).

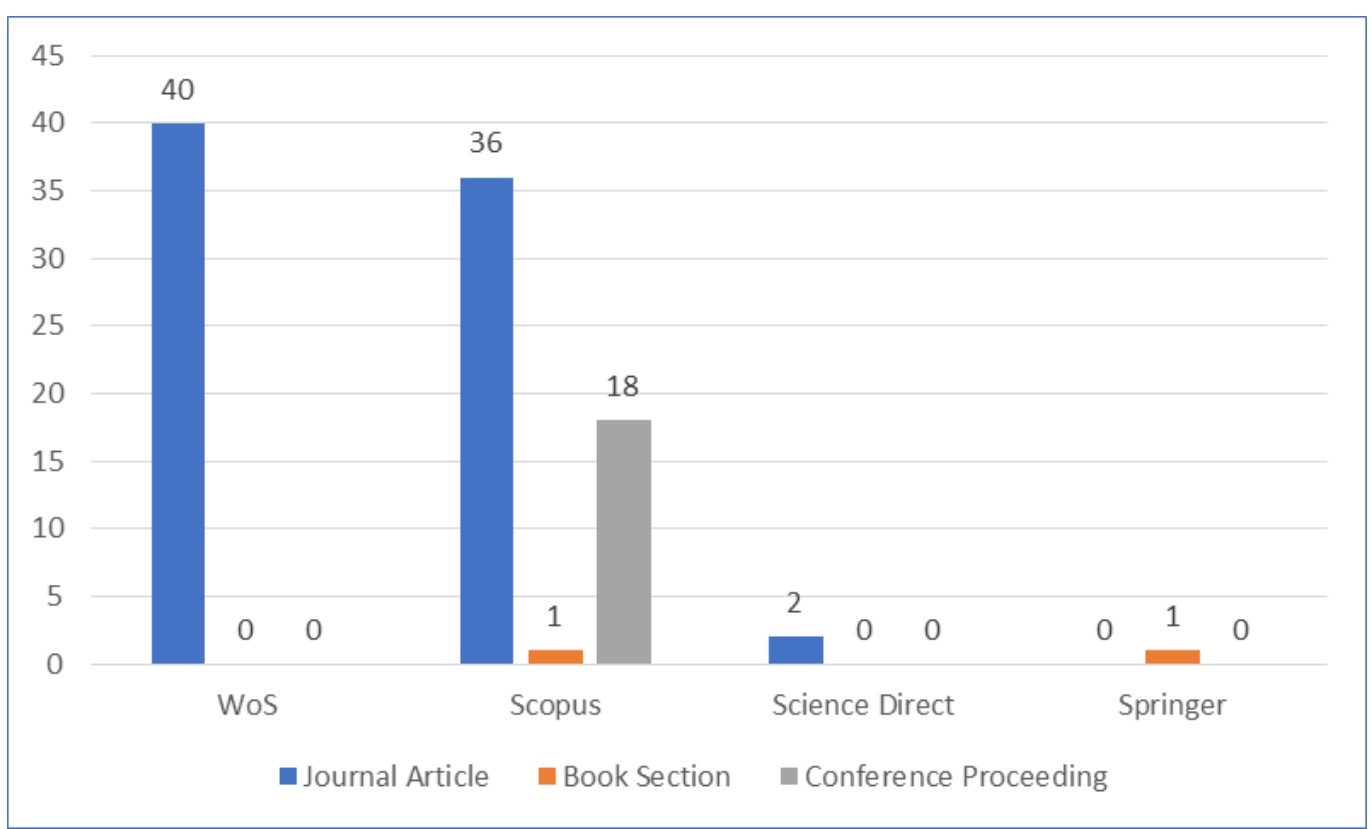

**Figure 6.** The number of jobs obtained by type from the databases.

Of 71 papers, 27 are doubly indexed in WoS and Scopus. There were 51 journal papers, 18 conference proceedings, and 2 book sections.

### 4.5. RQ5: How Many Papers on Transparency and E-Government in Electronic Procedures Address the Sustainable Development of Governments?

Based on RQ5, it cannot be determined that transparency and sustainability are unrelated because they may be implicit in some writings. However, the irrelevance of the papers to RQ5 does represent an opportunity for research associated with a sustainability and transparency framework that, to date (according to the selected papers), has not been adequately addressed or demonstrated. There are papers related to transparency and E-government, but they need to be addressed directly to sustainability according to the applied criteria.

## 5. Discussion

Electronic public procedures mediated by E-government systems play an essential role in national government programs since their purpose is to enable active transparency and to favor relations with suppliers efficiently [36]. Therefore, electronic public procurement systems are expected to contribute to increased transparency, which would bring about possible reductions in corruption [36,37], increased confidence in the procurement system, increased productivity of procurement activities as well as services, and attract new suppliers to do business with the government [16].

Despite these expectations, there is a perception that corruption would operate in the bidding of properties, services, and public tenders to recruit public sector personnel. Therefore, making electronic procedures transparent in the context of government bidding and contracting, whatever it may be, promotes monitoring by the public [23,38]. However, it is thought that as a culture of transparency consolidates, people will incorporate habitual practices in their behavior or professional performance that favor transparent processes [38].

Governments consider transparency an indispensable mechanism of public accountability and responsibility toward society [24]. It is used to make certain types of information available or to open specific decision-making processes to the public [24]. Transparency is related to accountability because if it does not exist, it is impossible to be accountable to citizens. After all, information and actions are deliberately hidden [39].

One of the main allies to increasing government processes' transparency and improving public service quality has been implementing and using information technologies in E-government [40]. In different countries, these initiatives are supported and regulated by laws on using information technologies in the public sector [3]. Thus, governments use electronic procurement and contracting systems to achieve a transparent process when acquiring properties or services and hiring personnel, often linked to procurement web portals [13]. Therefore, by making these processes transparent and giving citizens greater control, the aim is to improve trust in government.

The objective of this mapping was to draw an overview of the work on initiatives that have been used around transparency in electronic procurement in the context of E-government to identify which one or ones are associated with transparency and which ones effectively apply electronic procedures for transparency and thus enable sustainable development in E-government.

This section examines the relevant aspects of the reviewed studies. For this purpose, the analysis included studies that generated or used proposals for transparency in electronic government procedures, highlighting the methodologies used and their contributions. The studies analyzed correspond to the last five years to obtain an updated view of the problems, advances, challenges, and opportunities.

It is observed that the highest proportion of papers—20 papers, 28.57%—are oriented to analysis, comparisons, or reviews on public procurement issues. This is followed by 17 papers (24.29%) focusing on transparency and 10 papers (14.29%) on corruption. On

the other hand, the categories of maturity models, e-governance, ICTs, legal norms, and accountability show few papers, most of which are from the last two years (2021–2022).

We found a few papers proposing solutions oriented to transparency in electronic procurement.

In the study by Correa and Leal [29], the authors aimed to identify inflated prices in public purchases made by the federal government of Brazil, using unstructured data available on the transparency portal. The data were extracted and processed from this portal, and the last two years were considered for the research. Only medicines purchased by the Ministry of Health were studied [29]. The study's main result was a consolidated price base per medication to allow the identification of distortions in prices, facilitating the identification of cases that merit further investigation to unravel fraud to the treasury.

In the same context of public procurement, another study proposes a maturity model as a tool to measure the transparency of public tenders when government entities acquire software development [16]. The authors used a procedural methodology to support the design of maturity models along four dimensions: institutionalization, software procurement process, communication, and accountability. They defined a five-step model and tested it with genuine government buyers (suppliers) [16].

In subsequent work, the same authors propose an E-government new maturity model to measure the transparency of government entities that generate public procurement and personnel recruitment processes based on a literature review to determine the current state of research in the field [3]. Like the previous work, they proposed the following levels: (1) initial, (2) developing, (3) coordinating, (4) managing, (5) systematic, which allow the maturity levels of the transparency of e-procedures employed by state agencies to be evaluated in the following elements: institutionalization, property procurement process, service or consultancy procurement process, personnel recruitment process, communication, and accountability. We chose these levels and aspects to establish integration and innovation on a full-scale review of the specialized literature on this matter. A tool with these characteristics can be handy for measuring the degree of transparency in public entities, thus reducing corruption levels in their processes.

Concha et al. [14] also propose maturity models; the work proposes a maturity model in the electronic government procurement observatory context. It focuses both on the legal and institutional arrangements, as well as on the technical aspects of the portals; the model consists of two leverage domains, seven key domain areas, and twenty-five critical variables, which are rated using a weighted scoring system that produces quantitative indicators of portal capabilities and allows for direct comparisons [14].

Another exciting proposal consists of collecting data based on crowdsourcing to generate datasets of public procurement processes from unstructured information published on the web [30]. The results presented in this work are preliminary and require more exhaustive tests [30]. In this same logic of analysis of public information, in the study by Soylu et al. [31], the authors apply techniques to detect perturbations in an open and disparate dataset integrated into a knowledge graph, which includes data on tenders, companies, and expenditures, through a linked data-based platform. They present a set of guidelines for publishing high-quality procurement data to improve indicators. The authors present a set of recommendations to improve the way tenders are published. They use data related to public procurement, developing technologies and solutions for public and private sector buyers to use and adapt to be more transparent, make markets more competitive, and reduce waste and fraud [31].

In a recent paper [32], the authors developed a standard procurement data tool to promote transparency and compliance monitoring in the procurement process across government entities. The developed tool is expected to help government agencies promote transparency in the procurement process and monitor procurement activities to possibly eliminate mismanagement and corruption [32].

Although we learned about different initiatives that could contribute to E-government, only some papers propose solutions oriented to transparency in public procurement and e-procurement.

In two of the analyzed papers, the authors included the use of maturity models and accountability as critical dimensions to analyze and classify the transparency levels of a governmental entity. It is essential to point out that accountability is characteristic of representative governments since it allows voters to verify that their elected representatives comply with the declared objectives [41]. Verification through transparent procedures is a tool for accountability at the organizational level rather than at the individual level. Furthermore, accountability requires coordinated activities related to participation, openness, frequency, symmetry, proactiveness, synergy, and long-term interactions. These characteristics are, therefore, to be expected in transparent procedures.

In this context, countries' governments have established the concept of transparency in the various processes they carry out [42]. Transparency is understood as a concept that encompasses several ideas to curb corruption, including the establishment of simple decision-making processes, proper behavior of officials, public disclosure, integrity, accountability, and even democratic values [15].

There are no government initiatives or laws associated with transparency that would allow government agencies to effectively monitor and manage the implementation of such controls, let alone a methodological framework that promotes continuous improvement in this area [22,43]. This presents an opportunity to define a formal and systematic process to support organizations in defining, planning, implementing, and monitoring the transparency activities to which they should be subject [23].

Different governments have enacted laws that oblige public entities to make their procedures more transparent; however, the level of compliance with these regulations still needs to be higher [13,22]. One mechanism that can improve compliance with transparency laws and measure their impact is the generation of a maturity model. A maturity model is a map that guides the organization in the implementation of good practices, providing a starting point [44]; it constitutes an evolution of quality management practices. Initiatives can be found in software development, project management, knowledge management, process development, business process management, and supply chain management.

It is interesting to use maturity models for transparency in different government areas. Based on the analyzed papers [3,16,29–32,45], having a maturity model that allows measuring the compliance of any procedure will allow advances the improvement of transparency management (benefits of maturity models in practice). Furthermore, it is expected that the use of a diagnostic transparency methodology based on a maturity model will contribute to determining the degree of progress of a public organization concerning transparency initiatives, indicating at what level of transparency it can be classified.

The challenges to improving transparency in government require a regulatory framework to supervise how the different processes are carried out and to measure government entities' transparency levels. Thus, instruments and metrics play a crucial role in monitoring the change that is taking place. All this is fundamental for the sustainable development of transparency in E-government.

Sustainable development must be supported by transparency and E-government initiatives or policies, and knowing the number of studies focused on this becomes relevant for adopting sustainable development. However, although the works identified address transparency as a fundamental pillar in governments' sustainable development, none explicitly point out sustainability as a focus.

The United Nations have agreed to develop 17 sustainable development goals. Goal 16 explicitly addresses the institutional dimension, accountability, and transparency. Transparency has been raised as a value of democracy itself, and fair treatment, not only in terms of administration of justice but in equal and inclusive treatment by institutions, imposing an agenda and, with it, the imperative need to evaluate progress. However, a measure of progress cannot only be the implementation of norms in the sense of their replication. On the one hand, the normative value sub-stratum requires change, and on the other, the copying of norms does not necessarily imply legitimacy and, therefore, respect for them.

## 6. Limitations of the Study

Some limitations must be considered. As an example, this review could have included more relevant work.

Seventy-one papers published between 2012 and 2022 were identified that describe analyses, comparisons, or literature reviews regarding transparency in electronic procurement. Over the same period, five documents were identified proposing solutions oriented to transparency in electronic procurement. Furthermore, only one job we found includes a study or work related to transparency in electronic procurement for its subsequent application.

We have identified four types of threats to validity [21], considering that it is very difficult to guarantee absolute impartiality.

### 6.1. Construct Validity

In systematic mapping, threats to construct validity are relevant to the ranking of the selected studies. A search chain was performed using the WoS, Scopus, ScienceDirect, SpringerLink, and Wiley databases. According to search engine statistics, most research papers were on transparency and E-government in electronic procedures. To mitigate the risk of missing essential and related posts, we searched for related papers in state-of-the-art reports and surveys.

The search string used has a strong bias since there are keywords that need to be considered to obtain the most extensive set of studies possible.

Another innate bias is the initial filtering technique. When only examining titles and abstracts, some discarded papers may have included crucial information to answer the research questions.

From the definition of the inclusion and exclusion criteria, we left aside technical reports and works in languages other than English; therefore, some studies in other languages may have been relevant with proposals different from those found.

### 6.2. Internal Validity

Internal validity relates to the data extraction analysis process. At this stage, two authors identified the classification of the selected papers, and one author reviewed the results. Two reviewers had already applied subjectivity in applying the criteria above for study selection. Cohen's kappa index [28] was used to minimize possible research bias, obtaining a value of 0.90, which is acceptable.

### 6.3. External Validity

Concerning external validation, although the process was not evaluated by peers outside the study, applying a proven methodology accepted by the community ensures high-quality results. Likewise, the methodological foundations ensure that the proposal and its results are reliable and generalizable within the study domain.

### 6.4. Conclusion Validity

This type of threat refers to the possibility of identifying incorrect relationships between the results, which would lead to erroneous conclusions. To reduce the probability of this error, we designed a set of visual supports based on the results and statistical analysis performed to facilitate the visualization and analysis of the data and their relationships.

## 7. Conclusions and Future Works

This paper presented a systematic mapping of studies related to transparency and E-government in electronic procedures for an overall view of the solutions offered. For this, five research questions were designed, which were answered using different classifications of the selected works by the type of focus addressed, the kind of contribution, and the context.

The studies identified and classified the initiatives used around transparency and E-government in electronic procurement. One of the essential factors of those found was

the applicability of maturity models in government entities, as they imply the existence of a learning discipline within the organization.

At this point, maturity models are helpful, as they provide systematic frameworks to measure the performance of organizations in some regions of work, with greater strength in the technological field. A maturity model is a map that guides the organization in implementing good practices, providing a starting point. It describes an evolutionary improvement path, from the inconsistent processes to the most mature ones of the organization. It allows for assessing the state of development of an organization or business process, delineating improvement strategies to achieve planned objectives, and identifying the areas where the organization should focus on improving. Many authors use them to develop diagnoses and define measures of organizational progress.

The proposal of a maturity model implies the existence of a learning discipline within the organization. Further, a maturity model implies that within the framework of the processes, the actors recognize the relevance of developing specific actions or practices, establish standards and norms based on which they orient and evaluate their actions, practices, and strategies, and generate adjustments, changes, and innovations that make improvements in procedures, techniques, technologies, capabilities, competencies, and effectiveness feasible.

On the other hand, accountability is one of the attributes of the processes articulated around transparency. Accountability is situated within the framework of the search for representative governments, which includes control processes from citizens to elected officials through elections (vertical electoral accountability) and mechanisms of citizen participation (vertical social accountability). It also includes forms of control that operate in public administration, i.e., within the bureaucracy that implements public policies (internal vertical accountability, in terms of internal management control) and between that bureaucracy and those who provide services to governments (contractors, consultants, among others), who are considered a specific interest group.

In addition, the studies point out that one of the crucial challenges to increasing a government's transparency is to define a regulatory or legal framework that regulates its processes and allows measuring the levels of transparency or corruption in its different departments. Therefore, instruments and metrics are crucial in monitoring the expected change. With direct application in the organization, a model is essential for fundamental transparency in E-government.

In the fight against corruption, many countries now have transparency laws, i.e., open access to public information laws that allow citizens to access general information, including procurement information. In this scenario of openness, information and communication technologies play an essential role in facilitating the delivery of information to citizens to enable greater control over the actions of government and officials. Taking into account that, in some countries, public procurement information is provided in unstructured formats, this paper presented a data collection method based on crowdsourcing that allows the generation of public procurement datasets from information published on the web.

**Author Contributions:** Methodology, J.H. and M.D.; formal analysis, F.V. and M.D.; investigation, F.V.; resources, F.V.; writing, J.H. and J.A.-L.; application, M.D.; classification and categorization A.B. All authors have read and agreed to the published version of the manuscript.

**Funding:** This research was funded by Universidad de La Frontera, research direction, and the project DIUFRO DI22-0043.

**Institutional Review Board Statement:** Not applicable.

**Informed Consent Statement:** Not applicable.

**Conflicts of Interest:** The authors declare no conflict of interest.

## Appendix A

**Table A1.** Selected Papers.

| References | Title | Authors | Year | Citations |
|---|---|---|---|---|
| [46] | A comparative study on the present government procurement act and act for the promotion of private participation in infrastructure projects in Taiwan | Huang, P.-W. | 2016 | 2 |
| [47] | A Framework for the Adoption of Blockchain-Based e-Procurement Systems in the Public Sector: A Case Study of Nigeria | Akaba, T.I., Norta, A., Udokwu, C., Draheim, D. | 2020 | 29 |
| [48] | A Human Rights-based Approach to Combating Public Procurement Corruption in Africa | Mubangizi, J.C., Sewpersadh, P. | 2017 | 5 |
| [32] | A Procurement Data Standard Tool for State Universities and Colleges | Relucio, F.S., Dela Cruz, J.S. | 2020 | 0 |
| [49] | A study of organizational versus individual needs related to recruitment, deployment, and promotion of doctors working in the government health system in Odisha state, India | Kadam, S., Nallala, S., Zodpey, S., Pati, S., Hussain, M.A., Chauhan, A.S., Das, S., Martineau, T. | 2016 | 20 |
| [50] | A study on some aspects of E-Procurement in Indian organizations | Gupta, M., Narain, R. | 2012 | 2 |
| [51] | A toolkit for a prototype implementation of e-governance service system readiness assessment framework | Waseem, A.A., Ahmed Shaikh, Z., ur Rehman, A. | 2016 | 2 |
| [16] | A Transparency Maturity Model for Government Software Tenders | Hochstetter, J., Vairetti, C., Cares, C., Ojeda, M.G., Maldonado, S. | 2021 | 2 |
| [52] | Action research in procurement management; evidence from selected lower local government authorities in Tanzania | Rasheli, G.A. | 2017 | 6 |
| [53] | Advancing the E-Tendering Information System to Counter Corruption by Proposing Anti-Corruption SMART Tools | Delima, P.M., Dachyar, M. | 2020 | 3 |
| [54] | An empirical evaluation of the potential of public e-procurement to reduce corruption | Neupane, A., Soar, J., Vaidya, K. | 2014 | 61 |
| [55] | Analyzing the European union's tenders electronic daily: Possibilities and pitfalls | Prier, E., Prysmakova, P., McCue, C.P. | 2018 | 11 |
| [30] | Application of Crowdsourcing to Generate Datasets of Public Procurement Processes | Rey, A., Gomez, L., Lozada, A. | 2018 | 1 |
| [56] | Assessing The Role of Anti-Corruption Initiatives in Reducing Lobbyist Involvement In E-Procurement: A Case Study of Mardi | Ishak, M.W., Said, J. | 2015 | 16 |
| [3] | Assessing transparency in eGovernment Electronic Processes | Hochstetter, J., Diaz, J., Dieguez, M., Espinosa, R., Arango-Lopez, J., Cares, C. | 2022 | 1 |
| [57] | BlockchainAs a Service (BaaS) Framework for Government Funded Projects e-Tendering Process Administration and Quality Assurance using Smart Contracts | Hari Pranav, A., Latha, M., Ashwin, S., Chinnaiyan, R. | 2021 | 2 |
| [58] | Challenges in developing an E-government for good governance in North Sumatra | Siahaan, A.Y. | 2017 | 6 |

**Table A1.** *Cont.*

| References | Title | Authors | Year | Citations |
|---|---|---|---|---|
| [59] | Competition in public procurement in the fight against corruption: Analysis of an example of Ukraine | Psota, V., Chyzhevska, L., Osychka, O., Zaika, S., Koval, N. | 2020 | 3 |
| [60] | COST—The infrastructure transparency initiative- The disclosed data to sector reform | Hawkins, J., de Almeida Prado, M.D.G.F. | 2018 | 2 |
| [61] | Critical success factors for public-private partnership in the Afghanistan construction industry | Niazi, G.A., Painting, N. | 2018 | 8 |
| [31] | Data Quality Barriers for Transparency in Public Procurement | Soylu, A., Corcho, Ó., Elvesæter, B., Badenes-Olmedo, C., Yedro-Martínez, F., Kovacic, M., Posinkovic, M., Medvešček, M., Makgill, I., Taggart, C., Simperl, E., Lech, T.C., Roman, D. | 2022 | 1 |
| [62] | Degree of compliance with the Laws of Transparency, access, and good governance and of the Reuse of procurement data from the Spanish central government | Beltran-Orenes, P., Martinez-Pastor, E. | 2016 | 1 |
| [63] | Design of a Blockchain-based e-Tendering System: A Case Study in LPSE | Yutia, S.N., Rahardjo, B. | 2019 | 4 |
| [64] | Determinants of public-private partnership policies | Rosell, J., Saz-Carranza, A. | 2020 | 28 |
| [65] | Development Of Treasury Management Of Public Procurement: Problems And Prospects | Glazunova, I.V. | 2021 | 0 |
| [66] | Economic and Legal Problems of State and Municipal Procurement in the Russian Federation | Kozenko, Y.A., Perekrestova, L.V., Kurazova, D.A., Tereshkina, O.S., Golodova, O.A. | 2020 | 1 |
| [14] | E-Government procurement observatory, maturity model, and early measurements | Concha, G., Astudillo, H., Porrúa, M., Pimenta, C. | 2012 | 106 |
| [33] | Electronic government procurement implementation in India: A cross sectional study | Panda, P., Sahu, G.P. | 2015 | 16 |
| [35] | Electronic Government Procurement in the EU-Vietnam Free Trade Agreement: An Opportunity for Increased Transparency and Accountability? | Khorana, S., Kerr, W.A. | 2021 | 1 |
| [67] | E-procurement system success factors and their impact on transparency perceptions: Perspectives from the supplier side | Aminah, S., Ditari, Y., Kumaralalita, L., Hidayanto, A.N., Phusavat, K., Anussornnitisarn, P. | 2018 | 21 |
| [68] | Evolution Of Public Procurement Auctions In Russia | Melnikov, V.V.; Lukashenko, O.A. | 2019 | 9 |
| [69] | Foreign investment law and policy in Australia: a critical analysis | Bowman, M., Gilligan, G., O'Brien, J. | 2014 | 6 |
| [70] | Government procurement | Davies, A., Schefer, K.N. | 2015 | 0 |
| [71] | Hurting Pockets: A Case Study of Peru's Legal Obligations in Transparency and Justification of Public Expenditure in State Advertising | Calderon, A., Ascue, A., Dibos, E. | 2020 | 0 |

**Table A1.** *Cont.*

| References | Title | Authors | Year | Citations |
|---|---|---|---|---|
| [29] | Identification of overpricing in the purchase of medication by the Federal Government of Brazil, using text mining and clustering based on ontology | Correa, M.A.O.S., Galindo Leal, A. | 2018 | 2 |
| [72] | Impacts of the public procurement reform in Chile on the municipal level | Concha, G., Anrique, R. | 2012 | 2 |
| [73] | Improvement of Transparency through mining techniques for reclassification of texts: The case of Brazilian transparency portal | Almeida, G., Revoredo, K., Cappelli, C., Maciel, C. | 2018 | 2 |
| [74] | IT good governance: A case of the role of e-Procurement in Indonesia | Jonathan, K., Napitupulu, T.A., Sari, R. | 2018 | 4 |
| [75] | Land/Forest Acquisition After The Maluku Conflict, Its Impact On Ecosobling Rights Ownership Conflict Of Rights Owners And Ethics-Moral Implications Of Public Policies | Ruhulessin, J.C. | 2021 | 0 |
| [76] | Lights on the shadows of public procurement: Transparency as an antidote to corruption | Bauhr, M., Czibik, A., Licht, J.D., Fazekas, M. | 2020 | 49 |
| [77] | Measuring the efficiency of an entrepreneurial ecosystem at municipality level: does institutional transparency play a moderating role? | Riaz, M.F., Leitao, J., Cantner, U. | 2022 | 0 |
| [78] | Method for improvement of Transparency: Use of text mining techniques for reclassification of governmental expenditures records in Brazil | de Oliveira Almeida, G., Revoredo, K., Cappelli, C., Maciel, C. | 2021 | 0 |
| [79] | Nurturing domestic firms through public procurement: A comparison between Brazil and Japan | Sorte, W.F. | 2016 | 21 |
| [80] | Oil revenues, public procurement and armed conflict: A case study of a subnational government in Colombia | Rodriguez, J.D.G. | 2020 | 3 |
| [81] | Perceived benefits related to anti-corruption from e-tendering system in Nepal | Neupane, A., Soar, J., Vaidya, K. | 2012 | 5 |
| [82] | Preventing Procurement Fraud in E-purchasing for Indonesian Local Governments | Zahra, F., Abdullah, M.I., Kahar, A., Din, M., Nurfalah, N. | 2021 | 3 |
| [83] | Procurement contract management in the local government authorities (LGAs) in Tanzania: A transaction cost approach | Rasheli, G.A. | 2016 | 31 |
| [84] | Procurement fraud in the US Department of Defense Implications for contracting processes and internal controls | Rendon, J.M., Rendon, R.G. | 2016 | 77 |
| [85] | Procurement reform in the Philippines: The impact of elite capture and informal bureaucracy | Jones, D.S. | 2013 | 35 |
| [86] | Prototyping an intelligent contract based public procurement to fight corruption | Weingärtner, T., Batista, D., Köchli, S., Voutat, G. | 2021 | 6 |
| [87] | Public procurement in Malaysia: objectives and procurement principles | Abul Hassan, S.H.; Ismail, S.; Mutalib, H.A.A. | 2021 | 3 |

**Table A1.** *Cont.*

| References | Title | Authors | Year | Citations |
|---|---|---|---|---|
| [88] | Publishing construction contracts to improve efficiency and governance | Kenny, C. | 2012 | 29 |
| [89] | Regulating public procurement in Brazil, India, and China: Toward the regulatory-developmental State | Krizic, I. | 2021 | 7 |
| [90] | Requirement Specification in Government IT Procurement | Johansson, B., Lahtinen, M. | 2012 | 19 |
| [91] | Revenue transparency: global, not local solutions | Rees, P.J. | 2014 | 6 |
| [92] | Study of e-governance in India: a survey | Anand, D., Khemchandani, V. | 2019 | 2 |
| [93] | Tendering in assignment of the administrative contract: A comparison of Egyptian tender law and Saudi government tenders and procurement law | Alanzi, A.A. | 2021 | 0 |
| [94] | The determinant factors of individual performance from task technology fit and IS success model perspectives: A case of public procurement plan information system (SIRUP) | Diar, A.L., Sandhyaduhita, P.I., Budi, N.F.A. | 2019 | 6 |
| [34] | The E-Government and its Effects on Public Procurement | Amado, J.C. | 2020 | 0 |
| [95] | The impact of e-Procurement practice in Indonesia government: A Preliminary Study (The case of Electronic Procurement Service at Bekasi District) | Candra, S., Gunawan, F.E. | 2017 | 12 |
| [96] | The Interplay Of Incumbency, Political Dinasty And Corruption In Indonesia: Are Political Dynasties The Cause Of Corruption In Indonesia? | Purwaningsih, T., Widodo, B.E.C. | 2020 | 12 |
| [97] | TheyBuyForYou platform and knowledge graph: Expanding horizons in public procurement with open linked data | Soylu, A., Corcho, O., Elvesæter, B., Badenes-Olmedo, C., Blount, T., Yedro Martínez, F., Kovacic, M., Posinkovic, M., Makgill, I., Taggart, C., Simperl, E., Lech, T.C., Roman, D., Lehmann, J. | 2022 | 4 |
| [98] | Transforming information security governance in India (A SAP-LAP based case study of security, IT policy and e-governance) | Anand, R., Medhavi, S., Soni, V., Malhotra, C., Banwet, D.K. | 2018 | 13 |
| [99] | Transition to electronic procedures for government and municipal procurement in the course of national economy modernization | Goncharova, M.V., Baltutite, I.V., Guseinli, I.A.V. | 2019 | 3 |
| [100] | Transparency and Accountability Practices of Local Government Units in the Philippines: a Measurement from the Ground | Gabriel, A.G., Castillo, L.C. | 2020 | 8 |
| [101] | Transparency in public pharmaceutical sector: the key informants' perceptions from a developing country | Esfandiari, A., Yazdi-Feyzabadi, V., Zarei, L., Rashidian, A., Salari, H. | 2021 | 11 |
| [102] | Transparency level of the electronic procurement system in Malaysia | Ahmad, H., Abul Hassan, S.H., Ismail, S. | 2021 | 1 |

**Table A1.** *Cont.*

| References | Title | Authors | Year | Citations |
|:---:|:---:|:---:|:---:|:---:|
| [103] | Transparency of government revenues from the sale of natural resources: pursuing the international course through EITI | Poretti, P. | 2015 | 2 |
| [104] | TRANSPR—Transforming Public Accountability Through Blockchain Technology | Sriram, P.R., Subhashruthi, N.J., Muthu Manikandan, M., Gopalan, K., Kaushik, S. | 2021 | 1 |
| [105] | Using a DEA-cross efficiency approach in public procurement tenders | Falagario, M., Sciancalepore, F., Costantino, N., Pietroforte, R. | 2012 | 252 |
| [106] | Using Transparency Against Corruption in Public Procurement | Georgieva, I. | 2017 | 33 |

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
