# Peer review of "Transparency and E-Government in Electronic Public Procurement as Sustainable Development"

_sustainability, doi:10.3390/su15054672_

Round 1

Reviewer 1 Report

This article is of interest because it is devoted to a discussion of the corruption problem and ways to increase the financial transactions transparency, the use of which will increase the sustainability of the public administration system through e-government. The title of the article adequately reflects the content. In the abstract, the authors give the article essence, briefly describe the problem state, research methods, and results. Keywords correspond to the content of the article.

In the introduction, the authors provide a brief literature review on the article topic, indicate the goal and tasks of the study, and also give the article structure. The second section is devoted to the description of public electronic procedures. The third part is devoted to describing the method of conducting a systematic mapping study for scientific search and articles classification on the research topic. In the fourth section, the authors describe the selected articles, analyse the results. The fifth section contains a discussion of the results obtained. The sixth section includes study limitations. In the "Conclusions" section, the authors summarize the results obtained and draw conclusions on the work.

The article is prepared in accordance with the instructions for the authors, corresponds to the topic that it explores and publishes. In our opinion, the article corresponds to the topic “fighting corruption by increasing the transparency of financial transactions” and corresponds in type to the Preliminary Study.

Comment.

1.      The topic is interesting, but the literature review should be more thorough, because in this form the literature review is far from practical application. For example, it is not entirely clear how the authors narrowed down more than 437 articles to 110 and then to 71 articles for analysis in the second step. It is not entirely clear what the authors were guided by when highlighting the inclusion and exclusion criteria.

2.      The discussion is superficial. The authors were to discuss who benefits from the analysis and how it can be used for future research. A systematic review of the literature should synthesize the literature and not summarize the results of previous studies.

3.      In general, it is not clear how this systematic literature review, although with an attractive theme, reflects the scientific or practical content of the available research.

4.      In the second section, it makes no sense to single out subsection 2.1, since there are no other subsections (2.2, etc.)

5.      Titles of Table 3 and Table 4 and Figure 5 and Figure 6 in Spanish. Figure 5 comes after Figure 6.

6.      Sections 6.2 and 6.3 have the same title.

Reviewer 2 Report

This article is very interesting and well organized.

future research should be more deep and insight.

Reviewer 3 Report

Dear authors,

I really appreciate your paper and congratulate you for the performance.

However, I would like to suggest you to use more papers like references from articles already published in the last 1-3 years in journals indexed in SCOPUS or CA/WoS. In the same time, please use more references from the articles already published in MDPI journals in the last 1-3 years, related to your work.

I wish you all the best !

EG

Reviewer 4 Report

It is an interesting work. the methodology is correct and well applied. the results are relevant and help the development of the object of study. 

There is Spanish text on several pages, especially in the table designations (Table 3, paragraph two in Discussion ...) 

Reviewer 5 Report

The article is interesting. However, caution is advised: it is necessary to reinforce the introduction and make it a bit deeper. It is too descriptive. There is a lack of an authentic literature review or state of the question section. On the other hand, for example, Figure 5 and others have elements in Spanish (the title). A thorough revision of the writing is suggested. The object of study is not appropiate for Sustainability.

Round 2

Reviewer 5 Report

A phase two with own research is missing.

Round 3

Reviewer 5 Report

The article has improved.

Author Response

Thank you very much for your review and helpful comments throughout the review process. We have revised the English language again and corrected the detected flaws.